# A Novel Uremic Score Reflecting Accumulation of Specific Uremic Toxins More Precisely Predicts One-Year Mortality after Hemodialysis Commencement: A Retrospective Cohort Study

**DOI:** 10.3390/toxins12100634

**Published:** 2020-10-01

**Authors:** Yohei Arai, Shingo Shioji, Hiroyuki Tanaka, Daisuke Katagiri, Fumihiko Hinoshita

**Affiliations:** 1Department of Nephrology, National Center for Global Health and Medicine, Tokyo 162-8655, Japan; dkatagiri@hosp.ncgm.go.jp (D.K.); fhinoshi@hosp.ncgm.go.jp (F.H.); 2Department of Nephrology, Tokyo Medical and Dental University, Tokyo 113-8510, Japan; 3Department of Nephrology, Yokosuka Kyosai Hospital, Kanagawa 238-8558, Japan; infinity-stars@hotmail.co.jp (S.S.); htanaka@ykh.gr.jp (H.T.)

**Keywords:** end stage renal disease, uremic toxins, mortality, hemodialysis

## Abstract

Uremic toxins (UTs) generally accumulate in patients developing end-stage renal disease (ESRD). Although some kinds of UTs cause early death after starting hemodialysis (HD), it remains unknown whether the degree of excessive accumulation of various UTs is associated with worsening of prognosis. We retrospectively conducted this cohort study consisting of adult patients developing ESRD who initiated HD at the National Center for Global Health and Medicine from 2010 to 2019. We created a new uremic score, which was defined as the aggregate score of the following variables reflecting uremic state: elevated blood urea nitrogen, β2-microglobulin, and anion gap before starting HD. The primary outcome was early mortality within 1-year after HD commencement. The hazard ratio (HR) and 95% confidence interval (CI) for a one-point increase in uremic score was calculated with Cox proportional hazard models adjusted by baseline conditions. We included 230 participants, 16 of whom experienced the primary outcome of early mortality after HD commencement. Uremic score was significantly associated with the primary outcome (crude HR: 1.91, 95% CI 1.16–3.14; adjusted HR: 4.19, 95% CI 1.79–9.78). Our novel uremic score, reflecting accumulation of specific UTs, more precisely predicts early mortality after HD commencement.

## 1. Introduction

One of the main roles of the kidney is to excrete harmful metabolites and toxins produced by various metabolic activities in the body [1,2]. Although normal kidney with enough renal function provides efficient excretion of these uremic toxins (UTs), UTs generally accumulate in patients developing chronic kidney disease (CKD) [3]. Many UTs have been proven to perform unfavorable biological activity, resulting in poor prognosis, and causing uremic syndrome in patients developing end-stage renal disease (ESRD) [4]. Besides kidney transplantation, dialysis therapy has been established as a practical renal replacement therapy to remove a sufficient amount of small water-soluble and smaller mid-size UTs [5]. Although dialysis is generally initiated when subjective symptoms of uremic syndrome appear and are not treatable by other medications regardless of residual kidney function, there are only a few widely established objective indicators to diagnose uremic syndrome despite the variety of UTs.

Many UTs have been newly identified using integrative metabolomic and proteomic approaches in the past few years [4]. However, only a few UTs have an established role and practical application in a clinical setting. Among them, blood urea nitrogen (BUN) has been universally regarded as a surrogate for all accruing small water-soluble UTs, and an indicator of the need to start dialysis [6]. While higher pre-dialysis BUN has been proven to be related to high mortality after starting dialysis, it has long been demonstrated that BUN is likely not a major cause of uremic syndrome. [7]. In addition, BUN is influenced by glomerular filtration as well as other factors, including dietary protein intake, catabolic state, and volume status. Therefore, it is necessary to evaluate uremic state multilaterally using multiple indicators.

In the present study, we focused on β2-microglobulin (β2MG) and anion gap (AG) along with BUN, as easily measurable biomarkers reflecting the uremic state in patients with pre-dialysis ESRD in clinical practice. β2MG, which is a well-known representative marker of smaller mid-size UTs in patients undergoing maintenance dialysis, has recently been rediscovered as a novel filtration marker predicting CKD progression and mortality [8]. Additionally, an increased AG caused by retention of anionic UTs reflects the potential for progression of uremic syndrome in ESRD patients and has also been proven to be related to mortality around the commencement of dialysis [9,10]. Although each biomarker reflecting the accumulation of UTs has been proven to be related to poor prognosis in ESRD patients, it remains unknown whether the combination of these three biomarkers, reflecting accumulation of multiple specific UTs, shows a greater association with worsening prognosis. It is expected that a new composite indicator using specific biomarkers might be a more reliable indicator of prognosis in patients with ESRD. Moreover, few studies have verified the characteristics of each biomarker in ESRD patients.

In the present study, to evaluate uremic state multilaterally using a composite indicator, we assessed the patients’ uremic score, which was defined as the aggregate score of three variables related to uremia: elevated BUN, β2MG, and AG before starting HD. We evaluated whether uremic score was related to 1-year mortality after HD commencement in adult ESRD patients. Furthermore, we unraveled the characteristics of patients with elevated BUN, β2MG, and AG.

## 2. Results

### 2.1. Flowchart of Study Participation

Figure 1 shows the flowchart of patients in the present study. A total of 250 participants were included. Then, 20 patients were excluded due to the presence of comorbid unhealed cancer at the start of the observation period (2 patients), use of peritoneal dialysis (3 patients), or missing follow-up data (15 patients). Consequently, 230 participants were enrolled in this study. Eventually, 16 patients died (uremic score: 0, 2 patients (3%); uremic score: 1, 4 patients (5%); uremic score: 2, 6 patients (11%); and uremic score: 3, 4 patients (17%)) over a mean follow-up period of 347 days.

### 2.2. Baseline Characteristics

Table 1 demonstrates the baseline conditions of patients in this cohort. Patients showing a higher uremic score had higher phosphate and C-reactive protein (CRP) levels and lower estimated glomerular filtration rate (eGFR). Moreover, they had a lower frequency of cardiovascular disease and diabetes, and a lower history of malignancy. Furthermore, they had a lower frequency of nephrology care and medications. In addition, types of the previous malignant diseases were shown in Appendix A.

### 2.3. Association between Uremic Score and the Primary Outcome

Kaplan-Meier analysis indicated that patients with higher uremic scores had a lower survival rate (Figure 2). Evaluating the hazard ratio (HR) and 95% confidence interval (CI) for a one-point increase in uremic score was calculated with Cox proportional hazard models, uremic score was significantly related to the primary outcome (HR: 1.91, 95% CI 1.16–3.14; adjusted HR: 4.19, 95% CI 1.79–9.78) (Table 2). Furthermore, in a sensitivity analysis, adjusted uremic score was also associated with the primary outcome (Appendix A). In addition, crude hazard ratios of each baseline characteristic were demonstrated in Appendix A.

### 2.4. Underlying Characteristics of Patients with Elevated BUN, β2MG, and AG

In multivariate logistic regression analysis, all variables related to uremia were strongly associated with hyperphosphatemia (Table 3, Table 4 and Table 5). Elevated BUN positively correlated with hypotension and anemia, but negatively correlated with comorbid cardiovascular disease (Table 3). Elevated β2MG positively correlated with hypoalbuminemia and high CRP levels, but negatively correlated with use of statins and high eGFR at the start of hemodialysis (Table 4). Increase in AG positively correlated with a high CRP level, but negatively correlated with comorbid diabetes (Table 5).

## 3. Discussion

In the present study, we found that uremic score, defined as the aggregate score of three variables reflecting the uremic state, i.e., elevated BUN, β2MG, and AG before HD commencement, was related to 1-year mortality after HD commencement in adult ESRD patients. Moreover, elevated BUN mainly correlated with cardiovascular problems, including blood pressure and comorbid cardiovascular diseases, elevated β2MG levels mainly correlated with malnutrition and inflammation in patients with more advanced impairment of residual kidney function, and elevated AG mainly correlated with inflammation unaccompanied by malnutrition.

UTs are defined as many different substances that accumulate in the body with CKD progression, and which adversely affect biological functions [11]. Excessive accumulation of UTs causes various kinds of multiple organ dysfunction, with clinical features that constitute uremic syndrome. Therefore, appropriate evaluation of UTs is indispensable for future therapy and prevention of complications in ESRD patients [12]. In particular, it is important to determine the optimal timing of dialysis initiation without overlooking signs of UT accumulation. The optimal timing of dialysis initiation should be evaluated by not only eGFR, but also other multiple factors including renal failure symptoms, daily life activities, nutritional status, and uremic states [13,14]. Although there is no well-established method to evaluate all UTs accurately, the present study provides evidence supporting the importance of evaluating UTs multilaterally using selected indicators that reflect the uremic state.

BUN is the first UT to be identified and can be easily removed by dialysis, although the toxicity of BUN remains unclear. It is thought that uremic syndrome is related to the effects of other UTs, but not to BUN per se. On the other hand, BUN has more recently been found to directly increase reactive oxygen species and oxidative stress, leading to various types of tissue damage [15]. It has already been established that a high BUN level before starting HD is associated with a poor prognosis in patients with ESRD [16]. However, BUN is affected by various factors, such as production rate, catabolism, antidiuretic hormone release, and distribution volume, independent of decreased excretion by the kidney. In fact, in the present study, elevated BUN was strongly associated with not only hyperphosphatemia, which is a common biochemical abnormality found in patients with uremic state, but also hypotension and absence of cardiovascular comorbidities. These results suggest that BUN might not be an appropriate indicator reflecting accumulation of UTs in patients with cardiovascular problems.

β2MG, which is a well-known surrogate for smaller mid-size UTs, consists of non-variable light chains from the major histocompatibility complex (MHC) class I, and has recently been able to be efficiently removed from the blood due to advances in dialysis technology [17]. It has already been proven that pre-dialysis serum β2MG levels predict mortality, and the adequate removal of β2MG by HD is associated with improved prognosis in patients undergoing maintenance dialysis [8]. Moreover, β2MG has recently been regarded as a novel filtration marker of residual renal function also related to mortality, not only in patients undergoing maintenance dialysis, but also in pre-dialysis CKD patients [18]. A previous study demonstrated that β2MG is related to malnutrition and inflammation in patients undergoing maintenance HD [19]. In fact, in the present study, elevated β2MG was also associated with not only hyperphosphatemia, but also hypoalbuminemia and high CRP levels before starting HD. These results suggest that β2MG might be an important guide for deciding the timing of HD commencement in patients with more advanced ESRD complicated by protein-energy wasting, which is generally recognized as syndromes of inflammation, malnutrition, and muscle wasting in CKD patients [20].

AG is a simple indicator calculated by measuring anions and cations that is easily corrected by dialysis but is not a direct indicator reflecting the accumulation of UTs. However, an increase in AG is proven to be exacerbated by retention of unmeasured anionic UTs related to progression of uremia, and is related to 1-year mortality after HD commencement, particularly in the elderly [10]. While this previous study demonstrated that elderly patients with elevated AG usually have mobility impairment at the time of HD commencement, a major underlying pathology inducing mobility impairment and early mortality might be systemic inflammation, as indicated by the results of this study. In this study, unlike β2MG, elevated AG was significantly associated with high CRP level, but not hypoalbuminemia. These results suggest that AG might be a different indicator to determine the timing of starting HD from BUN and β2MG in terms of reflecting systemic inflammation in uremic patients.

In this study, we focused on β2MG and AG rather than BUN, as easily measurable indicators reflecting the uremic state in patients with ESRD in clinical practice that are corrected by commencement of dialysis. In fact, it is easy and useful to evaluate these three indicators in the clinical setting, compared with many other UTs, which can only be measured in the laboratory. However, it might be possible that there are other critical toxins that are not evaluated by using only these three indicators. Therefore, further research is needed to clarify the characteristics of each UT and establish a practical evaluation method assessing the accumulation of harmful UTs.

This study has some limitations. First, this study was conducted in a single-center. Therefore, the results, in this study, need to be validated by further studies conducted in multi-center. Second, confounding factors might have influenced variables reflecting the uremic state before starting HD, as the present study was an observational study. Third, this study likely had insufficient power because of the small sample size. Fourth, the measurement methods of serum bicarbonate level are not unified. In this study, the serum bicarbonate level was measured by blood gas analysis using either arterial, venous, or mixed venous blood. Fifth, AG is influenced by various factors such as serum protein concentration and the methodology used for measuring ion activities [21,22]. Since the serum sodium level was measured using the indirect potentiometry in the present study, it was particularly affected by abnormalities of total protein concentration, which were often developed in patients with ESRD. Therefore, we performed a sensitivity analysis using adjusted serum sodium level corrected for total protein concentration.

## 4. Conclusions

In conclusion, the novel uremic score defined as the aggregate score of each of the three indicators related to uremia: BUN, β2MG, and AG before HD commencement is related to 1-year mortality after HD commencement. While many indicators reflecting the accumulation of UTs in patients with ESRD have recently been identified as predictors of poor prognosis, each indicator has different individual characteristics. It is thus necessary to evaluate the uremic state multilaterally using composite indicators before starting HD.

## 5. Materials and Methods

The protocol of this study was approved by the ethical review board of the National Center for Global Health and Medicine (NCGM) [10,23]. The first edition of the research proposal was approved on November 9th, 2018 (approval number: NCGM-G-003076-00) and the second edition of the research proposal was approved on July 10th, 2020 (approval number: NCGM-G-003076-01). Because of the retrospective nature of the present study, the independent ethics committee at our hospital waived the need to get informed consent from participants. However, patients who declined to participate in our study were excluded. The research followed ethical principles laid down by the Declaration of Helsinki.

All patients who had complete data on BUN, AG, and β2MG, and started HD for ESRD at the NCGM from 2010 to 2019 were included in this study. Among them, participants with missing follow-up data, use of peritoneal dialysis, or comorbid unhealed cancer were excluded. Participants were followed up for a year after HD commencement or until the time of death, whichever came earlier. In addition, we decided the timing of HD commencement according to the section of “Hemodialysis Initiation for Maintenance Hemodialysis” in the Japanese Society for Dialysis Therapy Clinical Guideline [13]. The guideline described as following, “The judgment on the time to initiate hemodialysis is allowed when a residual renal function shows progressive deterioration and reaction to GFR < 15 mL/min/1.73 m^2^ in spite of sufficient optimal conservative treatment. However, the decision of starting hemodialysis should be determined based on a comprehensive assessment of renal failure symptoms, daily life activities, and nutritional status, which are not relievable without hemodialysis”.

The primary outcome of this study was 1-year mortality after HD commencement. To evaluate this primary outcome, the following data were extracted from medical records: demographic information including body mass index (BMI), clinical status, blood pressure, laboratory analyses, medications, and comorbidities. Values of the most recent laboratory data for the following items were recorded: BUN, AG, β2MG, eGFR, hemoglobin level, and serum levels of albumin, phosphate, and CRP. In this study, eGFR was evaluated by the estimation formula according to the Japanese Society of Nephrology [24]. In this study, serum sodium and chloride levels were measured using the indirect potentiometry and serum bicarbonate level was measured by blood gas analysis. Serum albumin level was evaluated by the modified bromcresol purple (BCP) method in this study [25]. An elevated AG level was evaluated as the change in AG corrected for albumin (ΔcAG), calculated with the following formula: ΔcAG = (serum sodium level − serum chloride level − serum bicarbonate level − 12) + 2.5 × (4.4 − serum albumin level). To evaluate the accumulation of various UTs, the uremic score, defined as the aggregate score of three clinical variables related to uremia, BUN > 100 mg/dL, ΔcAG > 5 mmol/L, and β2MG > 20 mg/L just before starting HD, was measured for each patient as follows: uremic score = 0, patients satisfied with not any three variables; uremic score = 1, patients satisfied any one variable; uremic score = 2, patients satisfied with any two variables; uremic score = 3, patients satisfied with all variables. For a sensitivity analysis, we redefined adjusted uremic score using adjusted ΔcAG calculated with the following equation: adjusted ΔcAG = (adjusted serum sodium level − serum chloride level − serum bicarbonate level − 12) + 2.5 × (4.4 − serum albumin level), adjusted serum sodium level = serum sodium level − 10.53 + (0.1316 × total protein concentration) [21]. We evaluated adjusted uremic score as the aggregate score of three indicators, BUN > 100 mg/dL, adjusted ΔcAG > 5 mmol/L, and β2MG > 20 mg/L. We included nephrology care for more than 6 months before HD commencement and the presence of a usable HD shunt (arteriovenous fistula or graft) as clinical status. Use of erythropoiesis-stimulating agents (ESA), statins, and renin-angiotensin system (RAS) inhibitors were investigated as medications. Comorbidities was assessed by histories of malignant diseases, diabetes, and cardiovascular disease including peripheral arterial disease (revascularization surgery, necrosis, or amputation), coronary artery disease (coronary revascularization, angina pectoris, or myocardial infarction), or stroke (cerebral and subarachnoid hemorrhage, transient ischemic attack, or cerebral infarction).

We divided patients into four groups based on their uremic scores. Baseline conditions of this cohort were demonstrated as mean ± standard deviation (SD) for continuous variables, and frequency and percentage for categorical variables. We compared the incidence of death within a year after HD commencement between these four groups using the Kaplan-Meier surviving curves and the log-rank test. The HR and 95% CI for a one-point increase in uremic score was calculated using Cox proportional hazard models. Adjusted HR was calculated by multivariate analyses using baseline covariates as following: Model 1, demographic data (age ≥ 75 years, sex, BMI < 20 kg/m^2^) and clinical status; Model 2, Model 1 and laboratory data (eGFR > 7 mL/min/1.73 m^2^, systolic blood pressure < 110 mmHg, hemoglobin < 7 g/dL, albumin < 3 g/dL, phosphate > 6 mg/dL); Model 3, Model 2 and comorbidities and medications. Moreover, we used logistic regression to unravel the characteristics of patients with elevated BUN, AG, or β2MG levels. Multivariate analysis was conducted using covariates that demonstrated a *p*-value of <0.05 in univariate analysis. We used SPSS software, version 25 (SPSS Inc., Chicago, IL, USA) for statistical analyses. In this study, statistically significance was determined by two-sided *p*-values < 0.05.

## Figures and Tables

**Figure 1 toxins-12-00634-f001:**
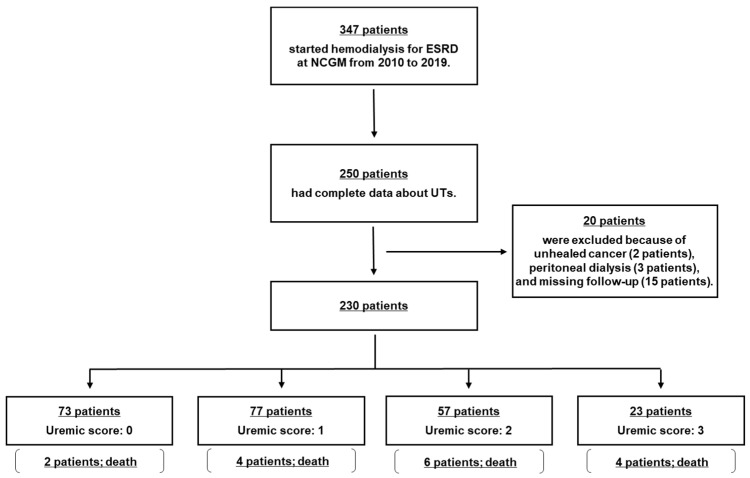
Flow chart of study participation. ESRD, end-stage renal disease; NCGM, National Center for Global Health and Medicine; UT, uremic toxins.

**Figure 2 toxins-12-00634-f002:**
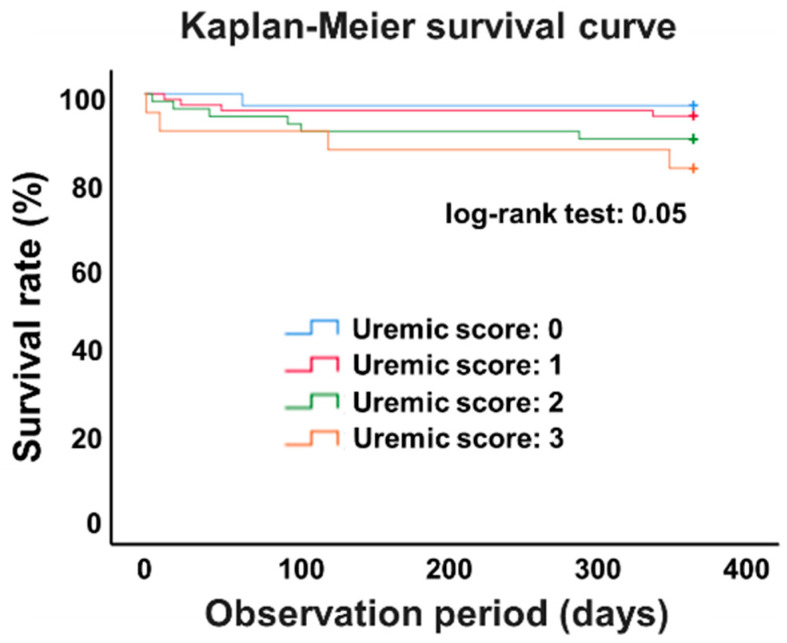
Kaplan-Meier curves of survival rates after hemodialysis commencement according to uremic score at baseline.

**Table 1 toxins-12-00634-t001:** Prognosis and baseline conditions of patients stratified into four groups based on uremic scores. Abbreviations: BMI, body mass index; eGFR, estimated glomerular filtration rate; CRP, C-reactive protein; BUN, blood urea nitrogen; ΔcAG, changes in anion gap corrected for albumin; β2MG, β2-microglobulin; RAS inhibitor, renin-angiotensin system inhibitor; ESA, erythropoiesis-stimulating agent. Continuous variables are shown as mean ± standard deviation. Categorical variables are shown as frequency and percentage.

Prognosis and Baseline Conditions	Overall	Score: 0	Score: 1	Score: 2	Score: 3
	(*n* = 230)	(*n* = 73)	(*n* = 77)	(*n* = 57)	(*n* = 23)
Prognosis					
Observation period (days)	347 ± 74	357 ± 49	352 ± 65	336 ± 90	322 ± 112
Death, *n* (%)	16 (7%)	2 (3%)	4 (5%)	6 (11%)	4 (17%)
Demographic characteristics					
Age (years)	68 ± 14	69 ± 12	68 ± 13	66 ± 15	67 ± 17
Male, *n* (%)	175 (76%)	56 (77%)	58 (75%)	43 (75%)	18 (78%)
BMI (kg/m^2^)	22.7 ± 4.7	23.2 ± 4.0	22.7 ± 4.7	22.9 ± 5.6	20.2 ± 4.1
Clinical status					
Presence of hemodialysis shunt	180 (78%)	65 (89%)	62 (81%)	38 (67%)	15 (65%)
Nephrology care (>6 months)	190 (83%)	64 (88%)	65 (84%)	43 (75%)	18 (78%)
Laboratory data					
eGFR (mL/min/1.73 m^2^)	5.4 ± 1.9	6.0 ± 1.7	5.5 ± 1.8	5.3 ± 1.9	3.8 ± 1.5
Systolic blood pressure (mmHg)	149 ± 21	146 ± 17	150 ± 23	149 ± 22	152 ± 22
Hemoglobin (g/dL)	8.8 ± 1.4	9.0 ± 1.2	8.9 ± 1.3	8.5 ± 1.6	8.8 ± 1.7
Albumin (g/dL)	3.1 ± 0.6	3.3 ± 0.6	3.0 ± 0.6	2.8 ± 0.7	3.2 ± 0.7
Phosphate (mg/dL)	5.9 ± 1.6	5.0 ± 0.9	5.7 ± 1.1	6.9 ± 1.6	7.5 ± 2.1
CRP (mg/dL)	1.7 ± 3.7	0.6 ± 1.5	1.3 ± 1.5	3.2 ± 5.4	2.9 ± 4.4
BUN (mg/dL)	89 ± 25	73 ± 15	85 ± 21	104 ± 25	119 ± 23
∆cAG (mmol/L)	5.7 ± 3.6	2.7 ± 1.8	5.4 ± 2.4	8.2 ± 3.1	10.2 ± 3.8
β2MG (mg/L)	18.8 ± 5.2	15.7 ± 2.2	18.2 ± 3.7	21.3 ± 6.7	24.3 ± 4.4
Comorbidities					
Diabetes, *n* (%)	136 (59%)	48 (66%)	50 (65%)	30 (53%)	8 (35%)
Cardiovascular disease, *n* (%)	82 (36%)	26 (36%)	34 (44%)	19 (33%)	3 (13%)
Malignant disease, *n* (%)	39 (17%)	8 (11%)	12 (16%)	12 (21%)	7 (30%)
Medication					
RAS inhibitor, *n* (%)	75 (33%)	31 (42%)	20 (26%)	19 (33%)	5 (22%)
Statin, *n* (%)	90 (39%)	36 (49%)	30 (39%)	20 (35%)	4 (17%)
ESA, *n* (%)	205 (89%)	68 (93%)	70 (91%)	48 (84%)	19 (83%)

**Table 2 toxins-12-00634-t002:** Hazard ratios of one point increase in the uremic score for 1-year mortality after hemodialysis commencement (*n* = 230). Multivariate Cox proportional hazards model was adjusted by baseline conditions: Model 1, demographic data (age ≥75 years, sex, BMI <20 kg/m^2^) and clinical status; Model 2, Model 1 plus laboratory data (eGFR >7 mL/min/1.73 m^2^, systolic blood pressure <110 mmHg, hemoglobin <7 g/dL, albumin <3 g/dL, phosphate >6 mg/dL); Model 3, Model 2 plus comorbidities and medications. Abbreviations: HR, hazard ratio; CI, confidence interval; *p*-value, two-sided probability value; BMI, body mass index; eGFR, estimated glomerular filtration rate.

Cox Proportional Hazard Model	HR (95% CI)	*p*-Value
One point increase in the uremic score		
Unadjusted model	1.91 (1.16, 3.14)	0.011
Model 1	1.70 (1.01, 2.83)	0.042
Model 2	2.44 (1.21, 4.95)	0.013
Model 3	4.19 (1.79, 9.78)	0.001

**Table 3 toxins-12-00634-t003:** Odds ratios for elevated BUN level in logistic regression analysis. Abbreviations: BUN, blood urea nitrogen; OR, odds ratio; CI, confidence interval; *p*-value, two-sided probability value; BMI, body mass index; eGFR, estimated glomerular filtration rate; CRP, C-reactive protein; RAS inhibitors, renin–angiotensin system inhibitors; ESA, erythropoiesis-stimulating agent. Multivariate analysis was conducted using covariates that demonstrated a *p*-value of <0.05 in univariate analysis. The hyphen in this table means no data available.

Logistic Regression Analysis	Univariate Analysis		Multivariate Analysis	
	OR (95% CI)	*p*-Value	OR (95% CI)	*p*-Value
Demography				
Age ≥ 75 years	1.38 (0.77–2.45)	0.273	-	-
Male gender	0.64 (0.34–1.22)	0.180	-	-
BMI < 20 kg/m^2^	1.65 (0.90–3.04)	0.102	-	-
Laboratory data				
eGFR > 7 mL/min/1.73 m^2^	0.95 (0.45–2.00)	0.896	-	-
Systolic blood pressure < 110 mmHg	11.9 (1.37–104)	0.025	29.1 (3.01–282)	0.004
Hemoglobin < 7 g/dL	3.30 (1.41–7.70)	0.006	2.88 (1.10–7.53)	0.031
Albumin < 3 g/dL	1.08 (0.61–1.91)	0.791	-	-
Phosphate > 6 mg/dL	5.12 (2.81–9.35)	<0.001	5.73 (3.00–10.9)	<0.001
CRP > 1 mg/dL	1.61 (0.88–2.97)	0.121	-	-
Comorbidities				
Diabetes	0.60 (0.34–1.06)	0.084	-	-
Cardiovascular disease	0.45 (0.24–0.85)	0.014	0.41 (0.20–0.85)	0.016
Malignant disease	1.96 (0.96–3.97)	0.062	-	-
Medications				
RAS inhibitor	0.98 (0.54–1.79)	0.963	-	-
Statin	0.60 (0.33–1.08)	0.092	-	-
ESA	0.43 (0.18–1.01)	0.054	-	-

**Table 4 toxins-12-00634-t004:** Odds ratios for elevated β2MG level in logistic regression analysis. Abbreviations: β2MG, β2-microglobulin; OR, odds ratio; CI, confidence interval; *p*-value, two-sided probability value; BMI, body mass index; eGFR, estimated glomerular filtration rate; CRP, C-reactive protein; RAS inhibitors, renin–angiotensin system inhibitors; ESA, erythropoiesis-stimulating agent. Multivariate analysis was conducted using covariates that demonstrated a *p*-value of <0.05 in univariate analysis. The hyphen in this table means no data available.

Logistic Regression Analysis	Univariate Analysis		Multivariate Analysis	
	OR (95% CI)	*p*-Value	OR (95% CI)	*p*-Value
Demography				
Age ≥ 75 years	0.98 (0.54–1.79)	0.958	-	-
Male gender	1.06 (0.54–2.10)	0.852	-	-
BMI < 20 kg/m^2^	2.00 (1.08–3.70)	0.027	1.84 (0.90–3.75)	0.090
Laboratory data				
eGFR > 7 mL/min/1.73 m^2^	0.23 (0.08–0.69)	0.008	0.20 (0.06–0.68)	0.010
Systolic blood pressure < 110 mmHg	1.27 (0.22–7.15)	0.780	-	-
Hemoglobin < 7 g/dL	0.98 (0.39–2.48)	0.976	-	-
Albumin < 3 g/dL	2.48 (1.38–4.46)	0.002	2.09 (1.04–4.19)	0.037
Phosphate > 6 mg/dL	3.23 (1.78–5.86)	<0.001	2.21 (1.13–4.30)	0.020
CRP > 1 mg/dL	2.93 (1.58–5.45)	0.001	2.30 (1.11–4.74)	0.024
Comorbidities				
Diabetes	0.67 (0.38–1.20)	0.188	-	-
Cardiovascular disease	0.73 (0.40–1.36)	0.333	-	-
Malignant disease	2.01 (0.98–4.12)	0.055	-	-
Medications				
RAS inhibitor	0.46 (0.23–0.91)	0.026	0.59 (0.28–1.26)	0.179
Statin	0.44 (0.23–0.84)	0.012	0.41 (0.20–0.84)	0.015
ESA	0.81 (0.33–2.00)	0.660	-	-

**Table 5 toxins-12-00634-t005:** Odds ratios for elevated ΔcAG level in logistic regression analysis. Abbreviations: ΔcAG, changes in anion gap corrected for albumin; OR, odds ratio; CI, confidence interval; *p*-value, two-sided probability value; BMI, body mass index; eGFR, estimated glomerular filtration rate; CRP, C-reactive protein; RAS inhibitors, renin–angiotensin system inhibitors; ESA, erythropoiesis-stimulating agent. Multivariate analysis was conducted using covariates that demonstrated a *p*-value of <0.05 in univariate analysis. The hyphen in this table means no data available.

Logistic Regression Analysis	Univariate Analysis		Multivariate Analysis	
	OR (95% CI)	*p*-Value	OR (95% CI)	*p*-Value
Demography				
Age ≥ 75 years	0.72 (0.42–1.24)	0.246	-	-
Male gender	1.42 (0.77–2.60)	0.258	-	-
BMI < 20 kg/m^2^	1.64 (0.91–2.97)	0.098		
Laboratory data				
eGFR > 7 mL/min/1.73 m^2^	0.64 (0.32–1.28)	0.215	-	-
Systolic blood pressure < 110 mmHg	0.85 (0.16–4.30)	0.846	-	-
Hemoglobin < 7 g/dL	3.01 (1.15–7.85)	0.024	2.17 (0.70–6.68)	0.177
Albumin < 3 g/dL	2.26 (1.31–3.91)	0.003	1.38 (0.73–2.60)	0.316
Phosphate > 6 mg/dL	4.75 (2.62–8.61)	<0.001	3.83 (2.03–7.22)	<0.001
CRP > 1 mg/dL	3.75 (1.96–7.15)	<0.001	2.85 (1.39–5.87)	0.004
Comorbidities				
Diabetes	0.50 (0.29–0.86)	0.013	0.48 (0.26–0.89)	0.021
Cardiovascular disease	0.98 (0.57–1.69)	0.954	-	-
Malignant disease	1.45 (0.72–2.94)	0.296	-	-
Medications				
RAS inhibitor	0.70 (0.40–1.22)	0.212	-	-
Statin	0.66 (0.39–1.13)	0.135	-	-
ESA	0.51 (0.21–1.24)	0.140	-	-

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
