# Peer review of "A Novel Uremic Score Reflecting Accumulation of Specific Uremic Toxins More Precisely Predicts One-Year Mortality after Hemodialysis Commencement: A Retrospective Cohort Study"

_toxins, 2020, doi:10.3390/toxins12100634_

Round 1

Reviewer 1 Report

The authors present a  novel uremic score reflecting accumulation of specific uremic toxins more precisely predicts early mortality after starting hemodialysis. In this score, anion gap and albumin "levels" (meant are activities for ions and concentration for albumin) play an major role.

This anion gap (AGP) is depicted by the authors as a "simple (!) measure.

In reality, in terminal renal insufficiency, this is not true! Aa the protein concentration is not standard (as admitted by the authors since they correct for it): there is a huge methodlogical problem caused by the atypical covolume of the solutes in dialysis patients. (ref: Stove V, Slabbinck A, Vanoverschelde L, et al. How to solve the underestimated problem of overestimated sodium results in the hypoproteinemic patient. Crit Care Med 2016;44: e83-e88 ). This problem is totally neglected by the authors. Their formula largely will depend on the methodology used for measuring ion activities (ref: Delanghe J. Management of electrolyte disorders: also the method matters!" Acta Clin Belg 2019; 74:2-6 ).

It is clear that the authors must mention their methodology used and should discuss the limitaions of their formula when other methodology is used!

Furthermore, also the albumin assay used should be mentioned. Correct measurement of albumin is extremely difficult in hemodialysis patient (ref: Delanghe JR, Speeckaert MM, Delanghe SE, De Buyzere ML. Albumin assays and clinical decision making in nephrotic syndrome patients. Kidn Int 2019;96: 248-249). Bromocresyl green? bromocresyl purple? immunochemical determination of albumin? this information is CRUCIAL for the reader.

The simple solutions offered by the authors are likely to be limited to their own centre (and hence less interesting for the readership of the journal). The limitations should at least be clearly stated in the manuscript.

minor issue: table 1: units for beta2 microglobuline concentration are missing

Author Response

We offer the reviewer real thanks for giving many meaningful advices. We hope that the revised paper will be more suitable for publication in Toxins.

The point-by-point responses to reviewer’s comments are:

  1. This anion gap (AGP) is depicted by the authors as a "simple (!) measure. In reality, in terminal renal insufficiency, this is not true! Aa the protein concentration is not standard (as admitted by the authors since they correct for it): there is a huge methodlogical problem caused by the atypical covolume of the solutes in dialysis patients. (ref: Stove V, Slabbinck A, Vanoverschelde L, et al. How to solve the underestimated problem of overestimated sodium results in the hypoproteinemic patient. Crit Care Med 2016;44: e83-e88 ). This problem is totally neglected by the authors. Their formula largely will depend on the methodology used for measuring ion activities (ref: Delanghe J. Management of electrolyte disorders: also the method matters!" Acta Clin Belg 2019; 74:2-6 ). It is clear that the authors must mention their methodology used and should discuss the limitaions of their formula when other methodology is used!

Thank you for your useful advise. We agree that anion gap is affected by various factors. Therefore, we additionally discussed this limitation in the revised manuscript. (Line 207-209)

In addition, we performed a sensitivity analysis using adjusted anion gap calculated using other methodology. We defined adjusted ΔcAG (change in anion gap corrected for albumin) using the following equation: adjusted ΔcAG = (adjusted serum sodium level – serum chloride level – serum bicarbonate level – 12) + 2.5 × (4.4 – serum albumin level), adjusted serum sodium level = serum sodium level – 10.53 + (0.1316 × total protein concentration) [Crit Care Med 2016; 44: e83-e88]. Then, we defined adjusted uremic score as the aggregate score of three clinical variables related to uremia, BUN >100 mg/dL, adjusted ΔcAG >5 mmol/L, and β2MG >20 mg/L. Finally, we demonstrated that adjusted uremic score was significantly associated with the primary outcome (hazard ratio: 1.68, 95% confidence interval 0.94-3.00; adjusted hazard ratio: 6.76, 95% confidence interval 1.27-35.9) (Supplementary Table S2). We added these descriptions in the revised manuscript. (Line 95-96, 252-258)

  1. Furthermore, also the albumin assay used should be mentioned. Correct measurement of albumin is extremely difficult in hemodialysis patient (ref: Delanghe JR, Speeckaert MM, Delanghe SE, De Buyzere ML. Albumin assays and clinical decision making in nephrotic syndrome patients. Kidn Int 2019;96: 248-249). Bromocresyl green? bromocresyl purple? immunochemical determination of albumin? this information is CRUCIAL for the reader.

Thank you for your valuable advise. We used the modified bromcresol purple (BCP) method in the present study [Clin Chim Acta 1999; 289(1-2): 69-78]. This description was added in our manuscript. (Line 244-245)

  1. The simple solutions offered by the authors are likely to be limited to their own centre (and hence less interesting for the readership of the journal). The limitations should at least be clearly stated in the manuscript.

Thank you for discussing important point. We had described this limitation in the Discussion section as following, “First, this was a single-center study. The results need to be validated by multicenter trials.” (Line 201-202)

  1. Minor issue: table 1: units for beta2 microglobuline concentration are missing

Thank you for your careful comment. We confirmed the correct spelling of “β2-microglobulin” in Table 1.

Reviewer 2 Report

The authors created a new uremic score using three indicators related to uremia: BUN, ß2-microglobuilin, and anion gap before starting HD and assessed the correlation of the uremic score and the mortality 1 year after starting HD. The new uremic score was significantly associated with the mortality, suggesting that the score more precisely predicts early mortality after starting HD.

Authors should describe how and when doctors decided the initiation of dialysis in these patients. It would be better to present the standard or the guidelines they used. Since eGFR has been widely used to decide to initiate dialysis, it would be necessary to compare eGFR with new uremic scores.

Authors selected three variables in this study. I think it would be better to compare each parameter including three variables and other characteristics with 1 year mortality.

Since the uremic scores were correlated with the mortality, it is interesting to know the change of uremic scores after initiating dialysis. Authors may compare three variables before starting dialysis and 1 year after dialysis, if possible.

In addition, were the uremic scores at 1 year after starting dialysis correlated with the mortality?

The data showed 17% of patients had malignant diseases. What kind of malignant diseases did the patients have?

Author Response

We offer the reviewer real thanks for giving many meaningful advices. We hope that the revised paper will be more suitable for publication in Toxins.

The point-by-point responses to reviewer’s comments are:

  1. Authors should describe how and when doctors decided the initiation of dialysis in these patients. It would be better to present the standard or the guidelines they used. Since eGFR has been widely used to decide to initiate dialysis, it would be necessary to compare eGFR with new uremic scores.

Thank you for your valuable advise. We followed the Japanese Society for Dialysis Therapy Clinical Guideline forHemodialysis Initiation for Maintenance Hemodialysis” [Ther Apher Dial 2015; 19: 93-107]. The guideline described as following, “The judgment on the time to initiate hemodialysis is allowed when a residual renal function shows progressive deterioration and reaction to GFR < 15 mL/min/1.73 m2 in spite of sufficient optimal conservative treatment. However, the decision of starting hemodialysis should be determined based on a comprehensive assessment of renal failure symptoms, daily life activities, and nutritional status, which are not relievable without hemodialysis”. Although eGFR has been widely used to decide to initiate dialysis, there could be other important factors to evaluate an optimal timing of dialysis initiation, including renal failure symptoms, daily life activities, nutritional status, and uremic states. In fact, a previous study in Japan demonstrated that the mean eGFR at dialysis initiation of patients without any symptoms was 4.74 mL/min/1.73 m2, and the mortality risk of patients without any symptoms who started hemodialysis with a high eGFR level was higher compared to that with a low eGFR level [Ther Apher Dial 2012; 16: 111-120]. Therefore, we considered the need to evaluate the timing of dialysis initiation multilaterally using multiple indicators including various uremic toxins other than eGFR. This description was added in our manuscript. (Line 151-153, 230-237)

  1. Authors selected three variables in this study. I think it would be better to compare each parameter including three variables and other characteristics with 1-year mortality.

Thank you for your useful advise. We demonstrated crude hazard ratios of each baseline characteristic in Supplementary Table S3.

  1. Since the uremic scores were correlated with the mortality, it is interesting to know the change of uremic scores after initiating dialysis. Authors may compare three variables before starting dialysis and 1 year after dialysis, if possible.
  2. In addition, were the uremic scores at 1 year after starting dialysis correlated with the mortality?

Thank you for your interesting suggestion. However, we couldn’t extract the uremic scores at 1 year after starting dialysis, because managements after the start of maintenance hemodialysis was mainly performed at other facilities. We think that this suggestion leads to future work for us.

  1. The data showed 17% of patients had malignant diseases. What kind of malignant diseases did the patients have?

Thank you for your meaningful advise. We demonstrated types of previous malignant diseases in Supplementary Table S1.

Round 2

Reviewer 1 Report

The paper has been improved

however , the reviewer is very dispappointed that the authors still do not report whether the sodium has been measured using INDIRECT or DIRECT potentiometry (please read refs 22 and 23 carefully).

The methodology for measuring electrolytes MUST be communicated.

In case of indirect potentiometry:the effect of low protein must be discussed

ref 22 Stove V, Slabbinck A, Vanoverschelde L, Hoste E, De Paepe P, Delanghe J. How to solve the underestimated problem of overestimated sodium results in the hypoproteinemic patient. Crit Care Med 2016;44: e83-e88  

ref 23 Delanghe J. Management of electrolyte disorders: also the method matters!" Acta Clin Belg 2019; 74:2-6  

Author Response

We offer the reviewer real thanks for giving repeated important advices. We hope that the revised paper will be more suitable for publication in Toxins.

The point-by-point responses to reviewer’s comments are:

  1. The reviewer is very dispappointed that the authors still do not report whether the sodium has been measured using INDIRECT or DIRECT potentiometry (please read refs 22 and 23 carefully). The methodology for measuring electrolytes MUST be communicated. In case of indirect potentiometry:the effect of low protein must be discussed.

Thank you very much for your repeated important advises. We added a sentence, “In the present study, serum sodium and chloride levels were measured using the indirect potentiometry and serum bicarbonate level was measured by blood gas analysis”, in the Method section (Line 247-248). Moreover, we added a limitation, “Because the serum sodium level was measured using the indirect potentiometry in the present study, it was particularly affected by abnormalities of total protein concentration which were often developed in patients with ESRD. Therefore, we performed a sensitivity analysis using adjusted serum sodium level corrected for total protein concentration.”, in the Discussion section (Line 209-212).

Reviewer 2 Report

Authors have performed the corrections and answered the comments in this version. Now I think this paper  improved very much.

Author Response

We offer the reviewer real thanks for giving repeated evaluations. I’m honored to receive your evaluation.